# Sex Differences in MASLD After Age 50: Presentation, Diagnosis, and Clinical Implications

**DOI:** 10.3390/biomedicines13092292

**Published:** 2025-09-18

**Authors:** Ilaria Milani, Maria Eugenia Parrotta, Luca Colangeli, Marianna Chinucci, Simonetta Palleschi, Barbara Rossi, Paolo Sbraccia, Alessandro Mantovani, Frida Leonetti, Valeria Guglielmi, Danila Capoccia

**Affiliations:** 1Department of Medico-Surgical Sciences and Biotechnologies, Faculty of Pharmacy and Medicine, University of Rome La Sapienza, 04100 Latina, Italy; chinuccimarianna@gmail.com (M.C.); frida.leonetti@uniroma1.it (F.L.); danila.capoccia@uniroma1.it (D.C.); 2Department of Systems Medicine, University of Rome Tor Vergata, 00161 Rome, Italy; mariaeugenia.parrotta@ptvonline.it (M.E.P.); luca.colangeli.ptv@gmail.com (L.C.); sbraccia@med.uniroma2.it (P.S.); valeria.guglielmi@uniroma2.it (V.G.); 3Obesity Medical Center, University Hospital Policlinico Tor Vergata, 00161 Rome, Italy; 4Department of Environment and Health, Istituto Superiore di Sanità, Viale Regina Elena 299, 00161 Rome, Italy; simonetta.palleschi@iss.it (S.P.); barbara.rossi@iss.it (B.R.); 5Department of Medicine, University of Verona, 37129 Verona, Italy; alessandro.mantovani@univr.it; 6Metabolic Diseases Research Unit, IRCCS Sacro Cuore-Don Calabria Hospital, 37024 Negrar di Valpolicella, Italy

**Keywords:** MASLD, obesity, sex-differences, type 2 diabetes, fibrosis, steatosis, FibroScan

## Abstract

**Background**: Age over 50, menopause, obesity and type 2 diabetes (T2D) are key risk factors for Metabolic dysfunction-associated steatotic liver disease (MASLD). This observational study aimed to assess sex differences in anthropometric and clinical profile, including non-invasive liver steatosis indices, in subjects with MASLD, obesity and/or T2D, aged ≥ 50 years. **Methods**: Anthropometric and clinical parameters, non-invasive indices for steatosis and fibrosis and FibroScan^®^ data were collected. **Results**: Among 213 patients (65.7% women, median age 63.0 years and mean Body Mass Index (BMI 34.9 kg/m^2^), men had higher body weight and waist circumference (WC), whereas women showed higher BMI and waist-to-height ratio (WHtR), and were more likely to exceed WC sex-specific and WHtR risk cut-offs. While transaminases values were higher in men, sex-specific cut-offs revealed that women more frequently exceeded these thresholds. No sex-differences were found for Fatty Liver Index (FLI), Fibrosis-4 (FIB-4) or FibroScan^®^, although higher rate of mild fibrosis in women. The diagnostic accuracy of FLI for detecting steatosis was significantly higher in men and unsatisfactory in women (Area Under the ROC Curve, AUC 0.863 vs. 0.655). **Conclusions**: While MASLD is more common in men, these results suggest that postmenopausal women with visceral obesity showed similar or worse liver and cardiometabolic profiles than men, despite appearing healthier based on standard clinical parameters. Notably, common markers like transaminases and the FLI were less accurate in detecting steatosis in women, underscoring the need for sex-specific diagnostic criteria and greater clinical attention to older women, particularly those with central obesity, to ensure early identification and management of MASLD.

## 1. Introduction

Metabolic dysfunction-associated steatotic liver disease (MASLD) is the most common chronic liver disease, affecting approximately 38.2% of adults worldwide [1,2,3]. It is diagnosed by the presence of liver steatosis with at least one cardiometabolic risk factor [4,5] and is associated with an increased risk of cardiovascular (CV) events and liver-related complications [4].

Early diagnosis of MASLD is essential, highlighting the need for the implementation of accessible, non-invasive and reliable diagnostic tools beyond standard reference imaging techniques [6,7]. Persistently elevated liver enzymes, such as alanine aminotransferase (ALT) and aspartate aminotransferase (AST), should prompt evaluation for metabolic dysfunction-associated steatohepatitis (MASH) and advanced fibrosis [4]. However, many patients with MASLD, MASH and advanced fibrosis may present with normal transaminase levels [4,8,9], underscoring the limited diagnostic accuracy of liver enzymes alone. To improve early detection, several non-invasive indices have been developed [10]. Among these, the Fatty Liver Index (FLI) is the only validated marker with strong diagnostic and prognostic value for identifying steatosis [6,11], while the Fibrosis-4 Index (FIB-4) is recommended for excluding advanced fibrosis [4,12]. Although clinical evaluation, laboratory tests, and ultrasound can help diagnose steatosis, they are not sufficient to assess its severity [7]. For this purpose, vibration-controlled transient elastography (VCTE) serves as a valuable second-line tool, providing a reliable estimate of both steatosis and fibrosis stage [4].

In the United States, the median age of people with MASLD was 50 years in 2015 [13], highlighting significant influence of aging on disease development [14,15]. Age-related changes in body composition are key contributors to the rising prevalence of MASLD and associated metabolic disorders [16]. Beyond age, sex and menopausal status also play a critical role [17,18,19], with notable sex-based differences in both the prevalence and severity of the disease. Men consistently exhibit a higher prevalence of MASLD [20,21,22] across all age groups, whereas in women, the risk increases with age [23], reaching or surpassing that of men after menopause [19,22,24,25,26].

While sex-differences in MASLD are increasingly acknowledged [17,19,24,27,28], sex-specific research, particularly studies using non-invasive diagnostic tools [29], in patients over 50 remains limited.

This observational study aims to evaluate sex differences in a population of MASLD over 50 years of age in anthropometric data, clinical profile (biochemical markers and non-invasive indicators of steatosis, fibrosis and cardiometabolic risk), and FibroScan^®^ assessments. We will also evaluate the influence of sex on the FLI prediction of liver steatosis (assessed by FibroScan^®^).

## 2. Materials and Methods

### 2.1. Study Design and Participants

A cross-sectional observational study was conducted in all consecutive patients aged ≥ 50 years and diagnosed with MASLD [4,30,31] at the Diabetes and Obesity Clinic of the Santa Maria Goretti Hospital in Latina, Italy, and the Obesity Medical Center of the Policlinico Tor Vergata Hospital in Rome, Italy from September 2024 to April 2025.

This is a secondary analysis of cohort data collected for the purpose of evaluating the efficacy of serum biomarkers in monitoring MASLD progression and metabolic complications over time. Considering that menopause typically occurs between the ages of 45–55 [32], with 51 being the generally accepted as the average age [33,34], the use of age > 50 as a proxy for menopausal status in women may help address this gap in the literature.

Exclusion criteria were as follows: (1) age younger than 50 years; (2) concomitant chronic liver diseases (e.g., viral, drug-induced liver injury, autoimmune hepatitis, hemochromatosis, primary sclerosing cholangitis, primary biliary cholangitis); (3) the presence of hepatocellular carcinoma or other malignancies; (4) excessive alcohol intake, defined as intake of >140 g/week for women, >210 g/week for men, detected with Alcohol Use Disorders Identification Test (AUDIT) questionnaire [4]; (5) clinical features suggestive of endocrinological disorders (Cushing syndrome, hypothyroidism, acromegaly, or Polycystic ovary syndrome); (6) pregnancy.

### 2.2. Anthropometric Assessment

Anthropometric measurements, including height (Ht), body weight (BW), and waist circumference (WC), were obtained during the clinical visit. Height and weight were measured with participants barefoot and wearing light clothing using a calibrated scale equipped with a standard stadiometer. BMI was calculated as weight in kilograms divided by the square of height in meters (kg/m^2^).

WC was measured at the level of the umbilicus with the patient standing upright, using a non-elastic anthropometric tape. Abdominal obesity was defined as a waist circumference ≥80 cm in women and ≥94 cm in men as a marker of visceral obesity and high risk of metabolic complications [4]. waist-to-height ratio (WHtR) was calculated by dividing WC (cm) by Ht (cm), with a fixed threshold of ≥0.50 used to indicate increased cardiometabolic risk [35].

### 2.3. Laboratory Analysis

After an overnight fast of at least 10 h, venous blood samples were obtained from all participants for biochemical analysis. Laboratory assessments included measurements of triglycerides (TG, mg/dL), total cholesterol (TC, mg/dL), high-density lipoprotein cholesterol (HDL-C, mg/dL), fasting plasma glucose (FPG, mg/dL), alanine aminotransferase (ALT, IU/L), aspartate aminotransferase (AST, IU/L), gamma-glutamyl transferase (GGT, IU/L), serum creatinine (Cr, mg/dL), and C-reactive protein (CRP, mg/L).

Quantitative estimation of biochemical parameters was performed in the central laboratory of the institute hospital participating to this study according to standard laboratory procedures using proprietary reagents on the fully automated biochemical analyzer Abbott Architect c16000 (Abbott Diagnostics, Abbott Park, IL, USA). Hemoglobin A1c (HbA1c) levels were determined using a high-performance liquid chromatography analyzer (Lifotronic Technology Co., Ltd., Shenzhen, China). Low-density lipoprotein cholesterol (LDL-C, mg/dL) was estimated using the Friedewald formula.

### 2.4. Non-Invasive Liver Disease Assessment (NILDAs)

MASLD was diagnosed in patients with evidence of hepatic steatosis detected by abdominal ultrasound or a FLI score ≥ 60 [30], in the presence of at least one cardiometabolic risk factor and in the absence of daily alcohol consumption (defined as <210 g/week for women and <140 g/week for men) or other liver diseases, in accordance with recent clinical recommendations [4,31].

The FLI was calculated using the established formula that includes BMI, WC, TG, and GGT levels [30]: [e^(0.953 × ln(TG) + 0.139 × BMI + 0.718 × ln(GGT) + 0.053 × WC − 15.745)]/[1 + e^(0.953 × ln(TG) + 0.139 × BMI + 0.718 × ln(GGT) + 0.053 × WC − 15.745)] × 100.

Although the FLI was originally validated to identify NAFLD, recent studies have shown that MASLD overlaps almost completely with the NAFLD population [36,37,38,39].

FIB-4 was determined by using the following formula: age [years] × AST [U/L]/(platelet count [10^9^/L] × √(ALT [U/L])) (where √ denotes the square root of ALT, measured in units per liter) [4].

For each patients the AST/ALT ratio was calculated and a value of 0.8 was considered to exclude advanced fibrosis [4].

All participants underwent VCTE with FibroScan^®^ Compact 530 after an overnight fast. With the patient in the supine position and the right arm abducted, the ultrasound probe was performed by professionally trained operators on the right lobe of the liver through intercostal spaces. The median value of the successful liver stiffness measurement (LSM) was expressed in kilopascals (kPa), while the median value of the successful Controlled Attenuation Parameter (CAP) score was expressed in decibels per meter (dB/m). Only LSM and CAP score data, acquired from at least 10 valid measurements, and IQR/median for both LSM and CAP score of <30% were considered reliable. Hepatic steatosis was simultaneously assessed by CAP and reported in decibels per meter (dB/m). Grades of steatosis were defined as follows:S0 (no steatosis): <248 dB/mS1 (mild): 248–267 dB/mS2 (moderate): 268–279 dB/mS3 (severe): ≥280 dB/m [4,40].

Fibrosis staging was based on the following LSM thresholds:F0: <7.9 kPaF1 (significant): ≥7.9–9.4 kPaF2 (advanced): 9.5–12.4 kPaF3–4 (cirrhosis): ≥12.5 kPa

These cut-offs are consistent with established guidelines for non-invasive assessment of liver fibrosis [4,41,42].

### 2.5. Cardiometabolic and Adipose Tissue Dysfunction Indices

Visceral adiposity index (VAI) was defined using WC, BMI, TG, and HDL-C according to the published sex-specific formula:Males: VAI = (WC [cm]/(39.68 + (1.88 × BMI [kg/m^2^]))) × (TG [mmol/L]/1.03) × (1.31/HDL [mmol/L])Females: VAI = (WC [cm]/(36.58 + (1.89 × BMI [kg/m^2^]))) × (TG [mmol/L]/0.81) × (1.52/HDL [mmol/L])

A VAI cut-off of 1.93 was used to identify adipose tissue dysfunction [43,44].

The TyG Index (TyG) was calculated as: Ln [TG (mg/dL) × fasting glucose (mg/dL)/2].

TyG-WHtR [45,46] was calculated as TyG index x WHtR.

### 2.6. Lifestyle and Other Variables

Data on age (years), sex, T2D and hypertension diagnosis, use of antihypertensive or antihyperlipidemic medication, cigarette smoking status were collected through oral interviews. Smoking status was classified into three categories: current smoker, former smoker, and never smoker. T2D was defined as fasting plasma glucose ≥ 126 mg/dL, 2-h plasma glucose ≥ 200 mg/dL after a 75-g oral glucose tolerance test, HbA1c ≥ 6.5%, clinical diagnosis by a clinician, and/or use of antidiabetic medication [47]. Hypertension was defined as systolic blood pressure (SBP) ≥ 140 mmHg and/or diastolic blood pressure (DBP) ≥ 90 mmHg or current use of antihypertensive medication.

Alcohol consumption was assessed using the validated Alcohol Use Disorders Identification Test (AUDIT) questionnaire. A score below 8 was considered indicative of low-risk drinking or abstinence [48]. In addition, alcohol consumption was quantified as the average number of standard drinks consumed per week.

Adherence to the Mediterranean diet was assessed using the adapted 14-item Mediterranean Diet Adherence Screener (MEDAS). A score of ≥9 was used as the threshold to categorize participants into high or low adherence groups [49].

### 2.7. Ethics Statement

The study was conducted in accordance with the Declaration of Helsinki [50], and approved by the “Comitato Etico Lazio 1”, the Ethics Committee of the Lazio Region (reference no. 7421, protocol code 0038/2024 and date of approval 10 January 2024). Written informed consent was obtained from all participants.

### 2.8. Statistical Analysis

Calculation of the sample size required to detect a medium effect size (Cohen’s d = 0.5) at a significance level of α = 0.05 and a desired power of 0.90 indicated a total sample size of 192 participants, including 64 in group 1 and 128 in group 2 (Power estimation was performed using G-Power version 3.1.9.4 software; Heinrich Heine University Düsseldorf, Düsseldorf, Germany).

The Kolmogorov-Smirnov test was used to assess the normality of the data distribution. Group comparisons were performed using Student’s *t*-test, Mann-Whitney U test, or chi-squared test, as appropriate. Continuous variables were expressed as mean ± standard deviation (SD) for normally distributed data or as median and interquartile range (IQR) for non-normally distributed data. Categorical variables were expressed as percentages.

A multivariate analysis of variance (MANOVA) was performed to examine the significant effect of sex on two dependent variables (CAP and LSM), adjusted for potential confounders.

Logistic regression model was used to evaluate the predictive ability of FLI for liver steatosis defined by CAP ≥ 248 dB/m, with adjustment for confounders and Receiver Operating Characteristic (ROC) curves were generated separately for men and women. To test the statistical difference between the two AUC curves, the DeLong test statistic was applied (using the free statistical software R 4.5.1).

Sex-stratified cross-tab analyses were performed to assess the association between FLI ≥ 60 and liver steatosis (CAP ≥ 248 dB/m), and sensitivity, specificity, positive and negative predictive values and accuracy of this cut-off were calculated for each sex.

In all analyses, a *p*-value < 0.05 was considered statistically significant. Statistical analyses were performed with SPSS software (version 21.0, IBM Corp, Armonk, NY, USA).

## 3. Results

A total of 213 patients were included in the analysis (Participants flow-diagram Appendix A). General and anthropometric characteristics of the study population are shown in Table 1. In the overall population, women accounted for 65.7% (140/213). The median age was 63.0 years (62.7–64.6), with a mean BMI of 34.9 kg/m^2^ (±5.5), median BW of 92.5 kg (92.9–97.7), mean WC of 114.2 (±12.7), and mean WHtR of 0.68 (±0.07). High prevalence of comorbidities such as T2D, hypertension and dyslipidemia have been observed in the overall population, without significant sex-differences. Higher significant percentage of men were active smokers (16.4% vs. 8.6%, *p* < 0.001) and former smokers (39.7% vs. 17.1%, *p* < 0.001), while no significant sex differences were observed for adherence to the Mediterranean diet assessed by MEDAS scores.

With regard to anthropometric parameters, body weight and WC were significantly higher in men compared to women (102.0 (99.5–107.5) vs. 88.5 (88.1–93.7) kg *p* < 0.001; 117.1 ± 12.4 vs. 112.7 ± 12.6 cm, *p* < 0.001). In contrast, WHtR was significantly higher in women than in men (0.69 ± 0.07 vs. 0.64 ± 0.12, *p* < 0.001), with the same trend for BMI (35.4 ± 5.6 vs. 33.8 ± 5.1 kg/m^2^, *p* = 0.051). Furthermore, the proportion of patients exceeding the sex-specific WC risk thresholds and those with a WHtR > 0.5 was significantly higher in women than in men (100% vs. 91.8%; 100% vs. 95.9%, *p* = 0.02, respectively).

**Table 1 biomedicines-13-02292-t001:** General and anthropometric characteristics of the study population.

	Overall (n = 213)	Men (n = 73)	Women (n = 140)	*p*-Value
Age (years)	63.0 (62.7–64.6)	63.0 (62.8–66.2)	63.0 (61.9–64.2)	0.10
Diabetes (%)	48.3	57.5	43.6	0.06
Hypertension (%)	76.5	76.7	76.4	0.97
Dyslipidemia (%)	63.4	63.0	63.6	0.88
Antihyperlipidemic drugs (%)	56.8	61.6	55.0	0.46
Current smokers (%)	11.3	16.4	8.6	0.001
Ex-smokers (%)	24.8	39.7	17.1	0.001
Medas ≥ 9 (%)	41.8	43.8	40.7	0.88
Weight (kg)	92.5 (92.9–97.7)	102.0 (99.5–107.5)	88.5 (88.1–93.7)	0.001
BMI (kg/m^2^)	34.9 ± 5.5	33.8 ± 5.1	35.4 ± 5.6	0.051
WC (cm)	114.2 ± 12.7	117.1 ± 12.4	112.7 ± 12.6	0.001
WC > 80 cm for women and WC > 94 cm for men (%)		91.8	100	0.02
WHtR	0.68 ± 0.07	0.64 ± 0.12	0.69 ± 0.07	0.001
WHtR > 0.5 (%)	98.6	95.9	100	0.02

Data are expressed as mean ± standard deviation (SD), number [n (%)] or median (interquartile range [IQR]). Medas: Mediterranean Diet Adherence Screener; BMI: Body Mass Index; WC: Waist circumference; WHtR: Waist-to-height ratio.

With regard to biochemical and clinical parameters (Table 2), women had significantly higher levels of total cholesterol (181.3 ± 41.1 vs. 162.5 ± 38.8, mg/dL, *p* = 0.001) and LDL-C (105.6 ± 36.9 vs. 88.6 ± 34.3, mg/dL, *p* = 0.02) than men. HDL-C levels were higher in women (52.0 (51.9–55.9) mg/dL) vs. 47.0 (45.5–56.7) mg/dL), although lower than the normal sex-specific HDL-cholesterol cut-offs. With regard to chronic low-grade inflammation, CRP concentrations were significantly higher in women than in men (0.35 (0.4–0.7) vs. 0.30 (0.2–0.6) mg/dL, *p* = 0.001).

The median values of ALT, AST and GGT were within the normal range in the whole population (21.0 (22.7–27.1); 21.0 (21.7–24.7); 24.0 (21.8–32.2)). However, ALT and GGT were significantly higher in men than in women (25.5 (24.4–32.4) vs. 19.0 (20.6–25.7) U/L, *p* = 0.001; 28.5 (28.9–45.5) vs. 22.0 (21.8–32.2) U/L, *p* = 0.02).

As regards NILDAs, FLI and FIB-4 did not show statistically significant differences between men and women (Table 3). However, the median AST/ALT ratio was significantly higher in women (1.1 (1.0–1.1) vs. 0.9 (0.8–1.0), *p* < 0.001), as was the proportion of women exceeding the clinical threshold of 0.8 (71.4% vs. 57.5%, *p* < 0.001). Furthermore, using the updated sex-specific cut-offs (≥33 U/L in men and ≥ 25 U/L in women for ALT and ≥30 U/L in men and ≥26 U/L in women for AST;) [4,51], the proportions of women with above-normal levels was higher than that of men for ALT and AST respectively (30.7% vs. 24.6%, *p* = 0.06; 24.3% vs. 19.2%, *p* = 0.09), although not significant. Among indicators of cardiometabolic risk and visceral adiposity, VAI was significantly higher in women than in men (1.97 (1.9–2.3) vs. 1.38 (1.4–2.1), *p* < 0.001). Similarly, the TyG-WHtR index was significantly higher in women (6.0 ± 0.8 vs. 5.6 ± 1.0, *p* = 0.02). A higher proportion of women than men had VAI values above the critical threshold of 1.93 (47.8% vs. 30.1%, *p* = 0.046).

The results obtained by VCTE (Table 4) showed that the median CAP value was significantly higher in men than in women (294.5 ± 50.9 vs. 277.4 ± 46.9 dB/m, *p* = 0.02), while no sex-differences were observed for LSM values. When analyzing the distribution of the degree of steatosis and fibrosis based on CAP and LSM thresholds, no significant sex-differences were found. However, a slight trend towards a higher prevalence of mild fibrosis (F1) was observed in women compared to men (16.5% vs. 8.2%, *p* = 0.07).

Multivariate analysis (Appendix A) confirmed the significant effect of sex and diabetes on CAP and LSM. Univariate analysis showed a significant effect of sex on CAP and diabetes on LSM. Other factors (age, smoking, dyslipidemia, hypertension) had no significant effect on CAP and/or LSM.

Logistic Regression Model used to evaluate the ability of FLI to predict liver steatosis as defined by CAP ≥ 248 dB/m (Table 5) reports the estimated coefficients and their confidence intervals, which were significantly higher in men compared to women. Furthermore, the model showed lower explanatory power in women, accounting for less variability in CAP values than in men, even after adjustment for confounders (diabetes, dyslipidemia, hypertension, smoking, age) (Appendix A).

The ROC analysis (Figure 1) showed that the diagnostic accuracy of the FLI for the detection of liver steatosis (CAP ≥ 248 dB/m) was significantly higher in men than in women. In women the Area Under the ROC Curve (AUC) was 0.655 (95% CI: 0.542–0.767, *p* = 0.014), whereas in men the AUC was 0.863 (95% CI: 0.731–0.995, *p* < 0.001). The difference between the two AUC was significant (*p* = 0.015).

In men, FLI ≥ 60 showed a sensitivity of 93.5%, a specificity of 70.0% with an overall diagnostic accuracy of 89.3% in predicting hepatic steatosis (i.e., CAP ≥ 248 dB/m). In women, while sensitivity remained high (90.2%), specificity dropped to 28.6% and negative predictive value (NPV) was only 47.1%, indicating a significant rate of false negatives (Table 6). The detailed contingency tables and chi-square test statistics are provided in Appendix A.

## 4. Discussion

Although MASLD is less prevalent in women [52,53], particularly during their reproductive years [24,54], its prevalence rises significantly after menopause [27], reaching levels comparable to those in men by around age 50, especially among individuals with obesity [17,55,56,57,58].

However, despite this convergence in prevalence, sex-related differences in clinical presentation, disease progression, and outcomes beyond midlife remain poorly defined.

Our results suggest that, after menopause, although women may appear metabolically healthier based on conventional clinical and biochemical parameters, a sex-specific analysis reveals a more unfavorable liver profile: despite mean transaminase levels were higher in men, applying sex-specific cut-offs (i.e., ALT ≥ 25 U/L and AST ≥ 26 U/L) [4,51] revealed a higher proportion of abnormal ALT and AST in women. Anyway, the proportion of individuals with elevated enzyme levels remained low in both sexes, consistent with existing evidence that plasma aminotransferases are weak indicators of MASLD [4,59], highlighting the need for additional diagnostic assessment.

NILDAs analysis also revealed similar mean FLI and FIB-4 levels in both sexes, and men had significantly higher mean CAP than women. On the other hand, when the population was stratified based on standard CAP and LSM cut-offs for steatosis and fibrosis, we found that the rate of early fibrosis (F1) was almost doubled in women. In addition, women had a higher AST/ALT ratio than men, which is recognized as a marker of liver inflammation [4]. This is supported by several studies reporting a higher AST/ALT ratio in women despite lower liver enzyme levels [60,61,62] or similar [63] to those in men. Moreover, a greater proportion of women also exceeded the AST/ALT ratio threshold associated with long-term liver complications, such as fibrosis [64], which is the strongest predictor of poor outcomes in MASLD [4].

Adipose tissue distribution changes with age and differs significantly between sexes [15,57], approximately doubling in men and quadrupling in women over time [65]. The age-related increase in VAT is reflected in higher WC, WHtR and BMI [66,67], particularly after menopause [68,69]. Notably, women tend to develop MASLD at higher total body fat percentages than men, with abdominal obesity [70], especially visceral obesity being an independent risk factor [69]. A large meta-analysis also reported consistently higher WHtR values in women with MASLD compared to men [71]. However, WC may underestimate visceral fat in postmenopausal women [69], as it primarily reflects SAT [72,73], or a mix of SAT and VAT [74], and appears to be a reliable marker of liver fat only in men [62,69]. Despite this limitation, the use of sex-specific WC thresholds improves risk stratification in women [75], as seen in our study.

In men, lower subcutaneous adipose tissue (SAT) levels contribute to greater ectopic fat deposition throughout life [25,70]. In contrast, women generally have higher SAT and benefit from a protective premenopausal hormonal environment [56,76]. During the menopausal transition, fat initially accumulates in SAT, followed by a gradual increase in total body fat and VAT [77], which helps delay the accumulation of fat in the liver [25].

The difference in SAT and VAT before and after menopause in women may also explain the differences in ALT and CRP levels observed in our study. Elevated liver enzymes, reflecting fat accumulation in the liver [78], are generally higher in men. However, both SAT and VAT are associated with markers of chronic inflammation, such as CRP [79]. In addition, the correlation between visceral fat and CRP appears to be stronger in women than in men [80,81,82]. Supporting this, studies on normal-weight subjects with obesity (NOW) have reported higher ALT in men, but higher CRP in women, compared to lean controls [83].

In our study, although higher absolute body weight and WC in men suggest a greater metabolic burden, sex-specific cut-offs show that women have increased total and visceral adiposity combined with more frequent early-stage liver fibrosis (F1). In addition, elevated cardiometabolic indices (e.g., VAI, TyG-WHtR) and more frequent exceeding of thresholds for these indices suggest greater metabolic dysregulation after menopause, likely driven by visceral adipose dysfunction and increased systemic inflammation, which may contribute to the observed sex differences after menopause [25].

In our population, a FLI was a good indicator of steatosis in men, but lacked specificity and negative predictive value in women. ROC analysis, also, highlights a low ability of FLI showed to detect hepatic steatosis in women, even after adjustment for several risk factors. On the other hand, the AASLD has reported limited accuracy of the CAP in quantifying hepatic steatosis [84], with measurement failures more common in female sex, probably due to differences in fat distribution [85]. These findings suggest that cut-off values for both CAP and LSM should be sex-specific, especially for postmenopausal women. Similarly, since sex-specific thresholds have been proposed for FLI components such as WC [86], the same consideration applies to the FLI score, which incorporates WC, and to the FIB-4 index, which has reduced accuracy in older adults [4]. However, neither index currently includes sex as a variable.

Although a FLI ≥ 60 in postmenopausal women shows limited accuracy in predicting steatosis as assessed by CAP, it still indicates an underlying metabolic risk especially when combined with other parameters associated with increased risk, such as WHtR, BMI, and WC. Some studies recommend substantially lower FLI cut-offs for women [11,86,87,88] up to 50% lower in those over 50, to improve detection accuracy in this group [11].

Overall, this supports the hypothesis that women with a positive FLI but no steatosis detected by CAP may be in the early stages of MASLD (S0-S1), that CAP may underestimate steatosis in women, and that fibrosis may develop even in the absence of CAP-detected liver steatosis.

This may explain the observed “sex paradox”, characterized by a non-linear trajectory of MASLD prevalence and progression across different ages and metabolic risk factors [19,25]. In fact, in our study, women showed more frequent early-stage fibrosis (F1) despite lower CAP values, whereas men had higher absolute CAP. This aligns with a large meta-analysis showing that although women have a lower overall prevalence of MASLD, they face a similar risk of MASH and an even higher risk of advanced fibrosis than men, especially after menopause [25]. Severe metabolic and inflammatory profiles were particularly evident in women at risk for MASH [89], and metabolic syndrome increased the risk of liver fibrosis by 2.7-fold, even without clear hepatic steatosis [90]. These findings are consistent with less favorable metabolic and inflammatory profiles observed in women, which may drive fibrosis progression despite lower CAP values.

The latest EASL-EASD-EASO guidelines identify men over 50, postmenopausal women, and individuals with multiple cardiometabolic risk factors as being at increased risk for progressive fibrosis and cirrhosis [4]. Therefore, this results highlight that women with MASLD, especially those over 50, should be evaluated for MASLD, MASH and fibrosis with equal or greater clinical vigilance compared to men [57].

These results must be interpreted taking into account the objectives and limitations. First, the identification of MASLD relied on non-invasive methods such as FLI and abdominal ultrasound rather than liver biopsy, which remains the gold standard for diagnosing liver fibrosis. However, FLI has shown predictive value for steatosis in various populations [30,91,92]. Second, the study was conducted exclusively in individuals with obesity, limiting the generalizability of the findings to MASLD patients without obesity. Third, the study is based on self-reported dietary and alcohol intake data obtained through questionnaires, which may be subject to recall bias. Fourth, because of its cross-sectional design, the study can detect associations but not causal relationships.

Key strengths include a well-characterized study population and the use of multiple non-invasive indices to assess steatosis, fibrosis, and cardiometabolic risk. Notably, the inclusion of VCTE provides a reliable assessment of hepatic steatosis and fibrosis. VCTE is a validated, cost-effective and non-invasive alternative to magnetic resonance elastography and liver biopsy, with strong correlation to histological findings in patients with NAFLD [93]. Importantly, this study specifically focused on individuals over 50 years of age, enabling a more accurate examination of sex-differences in a population where MASLD prevalence and progression are more comparable between men and women, thus addressing a critical gap in the existing literature.

## 5. Conclusions

While MASLD has historically been considered more prevalent in men, projections suggest a faster rate of increase among women by 2040 [94]. Similar to CV disease [95,96], MASLD in women often appears later, typically after menopause, and is associated with a greater burden of comorbidities and complications [25], higher mortality [97], increased cirrhosis prevalence [98], and a higher likelihood of being listed for liver transplantation [99]. Despite these risks, research focusing on middle-aged women remains limited, revealing a diagnostic gap similar to that in cardiovascular care, where the lack of sex-specific analyses has contributed to underdiagnosis and undertreatment in women [100].

The results of this study suggest that a more tailored approach is critical in postmenopausal women, especially those over 50, where increases in visceral fat may accelerate the progression of MASLD in women, even though healthier clinical appearances may delay diagnosis [25]. In particular, transaminase levels and FLI do not reliably identify liver steatosis in women compared to men, whereas the use of anthropometric height-adjusted measures and non-invasive markers of fibrosis and steatosis, together with the application of sex-specific cut-offs, may help to reveal a comparable or slightly higher burden of disease associated with visceral fat accumulation in women. Therefore, to prevent disease progression, especially after age 50, women may require closer clinical monitoring [25] using multiple non-invasive tools in addition to sex-specific cut-offs to enable early detection of these higher-risk female patients.

## Figures and Tables

**Figure 1 biomedicines-13-02292-f001:**
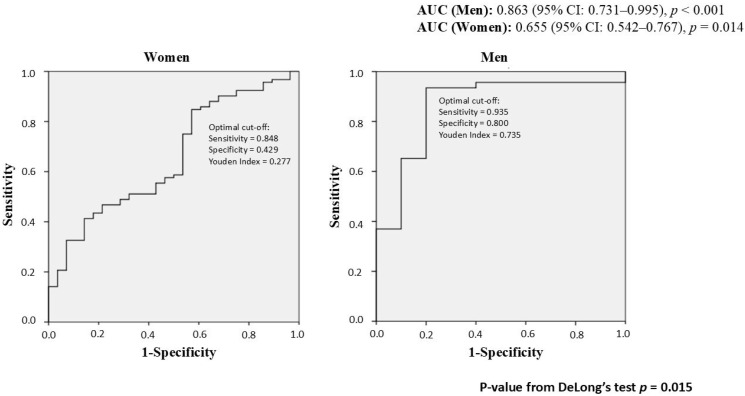
Sex-based analysis of Receiver Operating Characteristic (ROC) curves for FLI in predicting liver steatosis (CAP ≥ 248 dB/m). AUC: Area Under the ROC Curve.

**Table 2 biomedicines-13-02292-t002:** Biochemical and clinical parameters.

	Overall	Men	Women	*p*-Value
SBP (mmHg)	130.0 (129.1–133.5)	130.0 (126.3–134.4)	131.2 (129.2–134.4)	0.67
DBP (mmHg)	80.0 (78.5–81.4)	80.0 (78.2–82.8)	80.0 (77.7–81.6)	0.87
FBG (mg/dL)	99.0 (100.6–109.0)	104.0 (104.9–118.8)	96.0 (95.8–106.5)	0.001
HbA1c (%)	6.0 (5.9–6.8)	6.2 (6.0–6.7)	5.9 (5.8–7.0)	0.17
HbA1c (mmol/mol)	42.0 (42.9–46.3)	44.0 (42.9–52.2)	41.0 (41.1–44.8)	0.15
Total cholesterol (mg/dL)	175.1 ± 41.3	162.5 ± 38.8	181.3 ± 41.1	0.001
LDL cholesterol (mg/dL)	98.5 ± 36.8	88.6 ± 34.3	105.6 ± 36.9	0.02
HDL cholesterol (mg/dL)	50.0 (50.7–55.1)	47.0 (45.5–56.7)	52.0 (51.9–55.9)	0.001
TG (mg/dL)	122.5 ± 49.5	124.9 ± 60.4	119.7 ± 42.9	0.52
CRP (mg/dL)	0.30 (0.4–0.6)	0.30 (0.2–0.6)	0.35 (0.4–0.7)	0.001
Creatinine (mg/dL)	0.85 ± 0.2	1.01 ± 0.19	0.77 ± 0.18	0.001
ALT (U/L)	21.0 (22.7–27.1)	25.5 (24.4–32.4)	19.0 (20.6–25.7)	0.001
AST (U/L)	21.0 (21.7–24.7)	22.0 (22.2–27.9)	21.0 (20.6–23.6)	0.29
GGT (U/L)	24.0 (21.8–32.2)	28.5 (28.9–45.5)	22.0 (21.8–32.2)	0.02

Data are expressed as mean ± standard deviation (SD), number [n (%)] or median (interquartile range [IQR]). SBP: Systolic Blood Pressure; DBP: Diastolic Blood Pressure; FBG: Fasting Blood Glucose; HbA1c: Glycated Hemoglobin; LDL: Low-Density Lipoprotein; HDL: High-Density Lipoprotein; TG: Triglycerides; CRP: C-Reactive Protein; ALT: Alanine Aminotransferase; AST: Aspartate Aminotransferase; GGT: Gamma-Glutamyl Transferase.

**Table 3 biomedicines-13-02292-t003:** NILDAs, cardiometabolic and adipose tissue dysfunction indices.

	Overall	Men	Women	*p*-Value
FLI	88.1 (77.7–83.5)	92.5 (75.9–86.9)	86.0 (76.7–83.6)	0.31
FIB-4	1.45 (1.4–1.6)	1.43 (1.43–1.71)	1.45 (1.40–1.52)	0.75
AST/ALT	1.0 (1.0–1.1)	0.9 (0.8–1.0)	1.1 (1.0–1.1)	0.001
AST/ALT > 0.8 (%)	66.7	57.5	*71.4*	0.001
ALT ≥ 33 U/L for men and ≥25 U/L n (%)		24.6	30.7	0.06
AST ≥ 30 U/L for men and ≥26 U/L n (%)		19.2	24.3	0.09
TyG-WHtR	5.9 ± 0.9	5.6 ± 1.0	6.0 ± 0.8	0.02
VAI	1.82 (1.8–2.1)	1.38 (1.4–2.1)	1.97 (1.9–2.3)	0.001
VAI > 1.93 (%)	41.8	30.1	47.8	0.046

Data are expressed as mean ± standard deviation (SD), number [n (%)] or median (interquartile range [IQR]). FLI: Fatty Liver Index; FIB-4: Fibrosis-4 Index; AST/ALT: Alanine Aminotransferase to Aspartate Aminotransferase ratio; ALT: Alanine Aminotransferase; AST: Aspartate Aminotransferase; TyG-WHtR: TyG-waist-to-height ratio; VAI: Visceral Adiposity Index (cut-off > 1.93 indicates increased cardiometabolic risk).

**Table 4 biomedicines-13-02292-t004:** Sex-based analysis of VCTE parameters.

	Overall	Men	Women	*p*-Value
CAP (dB/m)	283.0 ± 48.9	294.5 ± 50.9	277.4 ± 46.9	0.02
CAP S0 (%)	22.5 (48/213)	16.4 (12/73)	25.7 (36/140)	0.29
CAP S1 (%)	13.6 (29/213)	17.8 (13/73)	11.4 (16/140)	0.29
CAP S2 (%)	11.7 (25/213)	10.9 (8/73)	12.1 (17/140)	0.82
CAP S3 (%)	53.0 (113/213)	57.5 (42/73)	50.7 (71/140)	0.32
LSM (kPa)	5.8 (5.9–6.5)	5.9 (5.9–6.9)	5.8 (5.7–6.4)	0.38
LSM F0 (%)	77.5 (165/213)	79.4 (58/73)	76.4 (107/140)	0.60
LSM F1 (%)	13.6 (29/213)	8.2 (6/73)	16.5 (23/140)	0.07
LSM F2 (%)	7.5 (16/213)	9.6 (7/73)	6.5 (9/140)	0.58
LSM F3–F4 (%)	1.4 (3/213)	2.7 (2/73)	0.7 (1/140)	0.55

Data are expressed as mean ± standard deviation (SD), number [n (%)] or median (interquartile range [IQR]). CAP: Controlled Attenuation Parameter; LSM: Liver Stiffness Measurement.

**Table 5 biomedicines-13-02292-t005:** Logistic regression model with FLI predicting CAP ≥ 248 dB/m.

		Beta Coefficient	Sig.	Exp(B)	95% CI per EXP(B)	
					Lower	Upper
Women	*FLI*	*0.026*	*0.018*	1.026	1.004	1.049
	Constant	−0.845	0.331	0.430		
Men	*FLI*	*0.064*	*0.001*	1.066	1.026	1.107
	Constant	−3.220	0.022	0.040		

FLI: Fatty Liver Index; Sig.: significance level; EXP(B): exponentiated coefficient (odds ratio); CI: confidence interval. For women: −2 Log Likelihood = 124.764, Cox & Snell R^2^ = 0.046, Nagelkerke R^2^ = 0.069. For men: −2 Log Likelihood = 38.039, Cox & Snell R^2^ = 0.228, Nagelkerke R^2^ = 0.375.

**Table 6 biomedicines-13-02292-t006:** Sex-based analysis of diagnostic performance of the FLI ≥ 60 in predicting liver steatosis (CAP ≥ 248 dB/m).

	Men	Women
Sensitivity (%)	93.5 (CI 95%: 86.4–99.6%)	90.2 (CI 95%: 82.6–95.3%)
Specificity (%)	70.0 (CI 95%: 41.6–98.4%)	28.6 (CI 95%: 13.2–48.7%)
Positive Predictive Value (%)	93.5 (CI 95%: 86.4–99.6%)	80.6 (CI 95%: 71.9–87.6%)
Negative Predictive Value (%)	70.0 (CI 95%: 41.6–98.4%)	47.1 (CI 95%: 23.0–72.2%)
Overall Accuracy (%)	89.3 (CI 95%: 81.2–97.4%)	75.8 (CI 95%: 67.1–83.1%)

Data are expressed as percentage (%). CI (Confidence Interval).

## Data Availability

The original contributions presented in this study are included in the article. Further inquiries can be directed to the corresponding author.

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
