# Peer review of "Sex Differences in MASLD After Age 50: Presentation, Diagnosis, and Clinical Implications"

_biomedicines, 2025, doi:10.3390/biomedicines13092292_

Round 1

Reviewer 1 Report

Comments and Suggestions for Authors

This is a cross-sectional study. The justification for using a non-invasive assessment is well presented and appropriately justified in the limited sensitivity of liver enzymes for detecting steatosis. The manuscript offers clinically relevant findings, but several methodological clarifications and analytic enhancements would improve rigor and interpretability.

Major comments

  1. Please clarify whether this is a secondary analysis of a prior cohort or an analysis of data routinely collected at the clinical site.
  2. The logistic regression should explicitly state that FLI was used to predict CAP. If the goal is a parsimonious diagnostic model, please describe why FLI was selected.
  3. The authors described several indices but ultimately emphasized FLI, despite no sex differences being observed in Table 3. Please provide the rationale for privileging FLI and show whether adding other clinically relevant predictors (e.g., age, BMI/waist, CRP, metabolic comorbidities, etc.) or considering sex interactions (given notable sex differences in Tables 1–3) improves AUC. If the authors have other steatosis indices available, consider comparative models or justify why they believe these indices do not add value.
  4. Interestingly, CRP levels were higher in women, while liver enzyme levels were higher in men. Please discuss possible biological or behavioral explanations for this phenomenon. Do they have predictive performance differing by sex?

Minor comments

Table 1: Rename the last column to “p-value” and remove the repeated “p=” in the rows.

Table 5: Change decimal separators from comma “,” to point “.” to match the format of other tables.

Tables 6 & Figure 1: These provide excellent diagnostic accuracy information.

In the regression table and caption, make the aim explicit: e.g., “Logistic regression model with FLI predicting CAP >= 248 dB/m .”

Consider adding a STROBE checklist and a simple participant flow diagram to strengthen transparency.

Author Response

Reviewer 1

Comments and Suggestions for Authors

This is a cross-sectional study. The justification for using a non-invasive assessment is well presented and appropriately justified in the limited sensitivity of liver enzymes for detecting steatosis. The manuscript offers clinically relevant findings, but several methodological clarifications and analytic enhancements would improve rigor and interpretability.

We sincerely thank the Reviewer for taking the time to review our manuscript and provide constructive feedback to improve it. We are very pleased and grateful to have been encouraged to revise the paper. Below are the reviewer's original comments and our corresponding responses, which are highlighted in yellow in the revised manuscript file.

Major comments

Comments 1: Please clarify whether this is a secondary analysis of a prior cohort or an analysis of data routinely collected at the clinical site.

Response 1: We thank the reviewer for this request. This is a secondary analysis of cohort data collected for the purpose of evaluating the efficacy of serum biomarkers in monitoring MASLD progression and metabolic complications over time. Since the inclusion criteria were age ≥ 50 years and several studies reported an inverse trend between sexes in MASLD after this age, we decided to perform a secondary analysis based on sex differences in this subpopulation (we have added this in lines 103-105).

Comments 2: The logistic regression should explicitly state that FLI was used to predict CAP. If the goal is a parsimonious diagnostic model, please describe why FLI was selected.

Response 2: We thank the reviewer for this suggestion and fully agree with the comment. In "2.8 Statistical analysis", we reported: "[...] Logistic regression model was used to evaluate the predictive ability of FLI for liver steatosis defined by CAP ≥ 248 dB/m (lines 243-244)". We also added the following sentence: "[...] Logistic regression model used to evaluate the ability of FLI to predict liver steatosis as defined by CAP ≥ 248 dB/m [...]" in the results (lines 334-335). The aim was to analyze the ability of a non-invasive index of liver steatosis to predict liver steatosis as assessed by FibroScan.

Comments 3: The authors described several indices but ultimately emphasized FLI, despite no sex differences being observed in Table 3. Please provide the rationale for privileging FLI and show whether adding other clinically relevant predictors (e.g., age, BMI/waist, CRP, metabolic comorbidities, etc.) or considering sex interactions (given notable sex differences in Tables 1–3) improves AUC. If the authors have other steatosis indices available, consider comparative models or justify why they believe these indices do not add value.

Response 3: We thank the reviewer for bringing this comment to our attention.

The FLI index is the best routine non-invasive validated tool to rule in/out liver steatosis in older adults in clinical practice [doi: 10.1016/j.cgh.2012. .12.031;10.3389/fnut.2025.1571487] and it has been reported the need to compare the prognostic value of CAP with FLI [DOI:10.1016/j.jhep.2013.12.018].  As reported in our manuscript (lines 159-161), several studies have shown a high degree of concordance of the FLI in the diagnosis of NAFLD, MAFLD, and MASLD, even when compared with the CAP [DOI: 10.1111/liv.12305]. The FLI includes anthropometric parameters such as BMI and WC, which change after menopause, and different cut-offs have been proposed, especially in women with MASLD over 50 years of age [doi: 10.1186/s13293-024-00617-z]. Because we found no sex differences in FLI levels (which we used to diagnose steatosis) in our cohort (Table 3), despite higher CAP levels in men, we decided to evaluate whether its predictive ability in detecting liver steatosis as assessed by CAP differed between sexes.

In addition, the adjusted logistic regression model for FLI vs. CAP (Table S3 in the Supplementary Appendix) showed that FLI retained a significant association even after adjustment for major confounders (age, diabetes, dyslipidemia, hypertension, smoking), with estimated coefficients significantly higher in men than in women. However, the addition of these covariates improved the model AUC in men (from 0.863 to 0.920, p<0.001), but not in women (from 0.655 to 0.680, p=0.005), confirming its greater predictive power in men, independent of the other factors considered, as reported in lines 335-337. Finally, we observed that the interactions between FLI and sex further reduced the predictive value of FLI in women (0.500) but not in men (0.863).

Comments 4: Interestingly, CRP levels were higher in women, while liver enzyme levels were higher in men. Please discuss possible biological or behavioral explanations for this phenomenon. Do they have predictive performance differing by sex?

Response 4: We thank the reviewer for this insightful comment and agree that this observation needs further explanation.

Elevated liver enzymes are an indirect indicator of fat accumulation in the liver [doi: 10.1186/s12933-015-0222-3], with generally higher levels in men than in women. This difference may be due to the lower amount of subcutaneous adipose tissue (SAT) in men, which facilitates more rapid deposition of ectopic fat, including liver fat. Conversely, in women, greater SAT and a more favorable premenopausal hormonal environment delay the accumulation of visceral fat (VAT), including liver fat. However, both SAT and VAT are associated with markers of chronic inflammation, including CRP [doi: 10.1161/CIRCULATIONAHA.107.710509], and a higher correlation between visceral fat and CRP has been reported in women than in men [doi: https://doi.org/10.1210/jc.2008-2406; doi: 10.1016/j.atherosclerosis.2005.04.011; doi: 10.3390/ijms20235981]. Supporting this, studies on normal-weight subjects with obesity (NOW) found higher ALT in men but higher CRP in women compared to lean controls [https://doi.org/10.1186/1475-2840-13-70].

Based on this evidence, we therefore interpret our findings as suggesting that the presence of larger SAT depots may attenuate local liver fat accumulation in women, despite being associated with higher levels of low-grade systemic inflammation. Furthermore, the use of lower sex-specific transaminase cut-offs may have helped to better detect the delayed liver fat accumulation in women, whereas the relative lack of SAT in men may promote earlier ectopic fat deposition in the liver, as reflected by higher transaminase levels, even in the presence of lower levels of chronic systemic inflammation.

Following the reviewer's suggestion, we have clarified this phenomenon in the text (lines 410-431).

Minor comments

Comments 1: Table 1: Rename the last column to “p-value” and remove the repeated “p=” in the rows.

Response 1: We thank the reviewer for this suggestion. We have changed the last column of Table 1 as suggested (we made this change for all tables for consistency).

Comments 2: Table 5: Change decimal separators from comma “,” to point “.” to match the format of other tables.

Response 2: We thank the reviewer for this accurate comment. We have changed the decimal separator from comma "," to period "." in table 5 as suggested.

Comments 3: Tables 6 & Figure 1: These provide excellent diagnostic accuracy information. In the regression table and caption, make the aim explicit: e.g., “Logistic regression model with FLI predicting CAP >= 248 dB/m”.

Response 3: We thank the reviewer for this comment and for the helpful suggestion. We have changed the title of table n.5 (line 343) and the relative capture (lines 334-336), as suggested.

Comments 4: Consider adding a STROBE checklist and a simple participant flow diagram to strengthen transparency.

Response 4: We agree with the reviewer's suggestion and have provided the participant flowchart (Fig S1 in the Supplementary Appendix) and the STROBE checklist (Strobe Checklist S4 in the Supplementary Appendix).

Reviewer 2 Report

Comments and Suggestions for Authors

This study investigates sex differences in MASLD among 213 patients aged ≥50 with obesity and/or T2D, finding men with higher body weight, WC, and CAP values, while women exhibited higher BMI, WHtR, rates of exceeding sex-specific risk thresholds, AST/ALT ratios, and mild fibrosis prevalence.

Major:

  1. The study includes 213 participants, with an imbalance between sexes (73 men vs. 140 women). Was a formal power calculation performed to determine if this sample is sufficient to detect meaningful sex differences, particularly in subgroups like those with T2D or hypertension? The small number of men may lead to underpowered analyses.
  2. The study emphasizes postmenopausal women as a key group, yet menopausal status is not explicitly confirmed. This is critical since the abstract and discussion attribute findings to menopause. How was menopause ascertained, and what proportion of women were truly postmenopausal? Without this, conclusions about estrogen's role may be speculative.
  3. MASLD diagnosis relied on ultrasound or FLI ≥60 plus cardiometabolic risks, per recent guidelines. However, ultrasound's sensitivity for mild steatosis is low, and FLI's validation was originally for NAFLD. Why not use CAP as the primary diagnostic tool for consistency?
  4. The conclusion that women have "similar or worse" profiles relies on sex-specific cut-offs, but absolute values (e.g., CAP higher in men) contradict this. How do these findings align with prior meta-analyses showing male predominance in MASLD progression? The "sex paradox" is mentioned but not deeply explored—please elaborate on the underlying mechanisms.
  5. The use of t-tests, Mann-Whitney, and chi-square is appropriate, but multivariable adjustments are lacking in key comparisons (e.g., sex differences in CAP or LSM). Logistic regression for FLI vs. CAP is good, but why no adjustment for confounders? ROC analyses show sex differences in AUC, but confidence intervals overlap slightly—did you test for significant AUC differences?

Minor:

  1. As for the title, the authors mention "progression," but the study is cross-sectional—no progression data. Please revise to "presentation" or "characteristics."
  2. Please clarify "unsatisfactory" FLI in women—specify AUC values in the abstract.
  3. Tables 1-4 show some inconsistencies in reporting (e.g., age as median in text but mean in Table 1). p-values: Some borderline (e.g., BMI p=0.05)—report exact values. 4. Figure 1: Add sensitivity/specificity at optimal cut-offs.
  4. In Table 5, please clarify "B" as the beta coefficient.
  5. All tables should add footnotes for abbreviations (e.g., PAS=PAD). And please ensure consistent units (e.g., HbA1c in % and mmol/mol).

Author Response

Reviewer 2

Comments and Suggestions for Authors

This study investigates sex differences in MASLD among 213 patients aged ≥50 with obesity and/or T2D, finding men with higher body weight, WC, and CAP values, while women exhibited higher BMI, WHtR, rates of exceeding sex-specific risk thresholds, AST/ALT ratios, and mild fibrosis prevalence.

Author's reply to reviewer's report:

We sincerely thank the reviewer for his careful reading of the manuscript and his constructive comments. We have used the valuable comments to improve and clarify our manuscript. We have addressed all concerns in the revised manuscript, and below we provide a point-by-point response to the reviewer's comments, which are highlighted in yellow in the revised manuscript.

Major comments:

Comments 1: The study includes 213 participants, with an imbalance between sexes (73 men vs. 140 women). Was a formal power calculation performed to determine if this sample is sufficient to detect meaningful sex differences, particularly in subgroups like those with T2D or hypertension? The small number of men may lead to underpowered analyses.

Response 1: We thank the reviewer for raising this important issue. Yes, a formal power calculation suggested that to detect a medium effect size (Cohen's d = 0.5) at a significance level of α = 0.05 and a desired power of 0.90, a total sample size of 192 participants was required, including 64 in group 1 and 128 in group 2 (power estimation was performed using G-Power version 3.1.9.4 software). Accordingly, we added information in the text (lines 230-233). Subgroup analysis by T2D or hypertension was not the aim of our study. However, because the percentage of diabetes was slightly higher in men than in women, we performed a subgroup analysis based on the presence of T2D, but no significant differences were found from our results. Therefore, we did not report it in the paper.

Comments 2: The study emphasizes postmenopausal women as a key group, yet menopausal status is not explicitly confirmed. This is critical since the abstract and discussion attribute findings to menopause. How was menopause ascertained, and what proportion of women were truly postmenopausal? Without this, conclusions about estrogen's role may be speculative.

Response 2: We thank the reviewer for bringing this point to our attention. We certainly agree with this observation. Since it has been reported in the literature that the international average age of natural menopause is 45-55 years [doi: 10.1016/S2214-109X(20)30215-1], and 51 is the average age accepted as the reference age for menopause in most developed countries [PMID: 29904254; doi: 10. 2147/IJWH.S228594], we use age >50 years as representative of postmenopausal status in women (consistent with their median age of 63 [61.9-64.2] years). We have added this information in the revised version of the manuscript (lines 105-108).

Comments 3: MASLD diagnosis relied on ultrasound or FLI ≥60 plus cardiometabolic risks, per recent guidelines. However, ultrasound's sensitivity for mild steatosis is low, and FLI's validation was originally for NAFLD. Why not use CAP as the primary diagnostic tool for consistency?

Response 3: We thank the reviewer for this careful consideration. FLI index is the best routine non-invasive validated tool to rule in/out liver steatosis in older adults [doi: 10.1016/j.cgh.2012.12.031; 10.3389/fnut.2025.1571487]. CAP cut-offs are not clearly defined and may overestimate or underestimate the prevalence of liver steatosis in published studies when considered alone [doi: 10.1007/s10396-021-01106-1].  A recent meta-analysis recognized that its value in classifying MASLD patients is in doubt, given that it is influenced by several factors, including sex [DOI: 10.1016/S2468-1253(20)30357-5]. As reported in our manuscript (lines 159-161), several studies have suggested a high degree of concordance of the FLI in the diagnosis of NAFLD, MAFLD, and MASLD, even when compared with the CAP [DOI: 10.1111/liv.12305]. Furthermore, when the availability of FibroScan in routine clinical practice is limited, FLI represents a simple and cost-effective tool based on commonly available metabolic parameters in a low-income and resource-limited health care setting, to be followed by more advanced investigations such as FibroScan as a second tool. We therefore decided to use FLI to diagnose steatosis and to assess whether CAP values were consistent with the resulting diagnosis in both sexes.

Comments 4: The conclusion that women have "similar or worse" profiles relies on sex-specific cut-offs, but absolute values (e.g., CAP higher in men) contradict this. How do these findings align with prior meta-analyses showing male predominance in MASLD progression? The "sex paradox" is mentioned but not deeply explored—please elaborate on the underlying mechanisms.

Response 4: We thank the reviewer for this very helpful comment. We have provided a detailed explanation of this mechanism in the discussion in hopes of clarifying it (lines 398-401 and 410-467).

Comments 5: The use of t-tests, Mann-Whitney, and chi-square is appropriate, but multivariable adjustments are lacking in key comparisons (e.g., sex differences in CAP or LSM). Logistic regression for FLI vs. CAP is good, but why no adjustment for confounders? ROC analyses show sex differences in AUC, but confidence intervals overlap slightly—did you test for significant AUC differences?

Response 5: We thank the reviewer for this thoughtful comment. A multivariate analysis of variance (MANOVA) was performed to examine the effect of sex on the dependent variables (CAP and LSM), adjusted for potential confounders (Table S2 in the Supplementary Appendix).

Table S2. Multivariate analysis of variance (MANOVA) on the dependent variables (CAP and LSM). 

Effect

Test Type

Dependent Variable

F

df

p-value

Partial η²

Sex

Multivariate

3.852

2

0.023

0.036

Univariate

CAP

6.580

1

0.011

0.031

Univariate

LSM

2.036

1

0.155

0.010

Diabetes

Multivariate

4.650

2

0.011

0.043

Univariate

CAP

1.251

1

0.265

0.006

Univariate

LSM

8.844

1

0.003

0.041

F-values (F), degrees of freedom (df), significance levels (p-value), and partial eta-squared (η²) are reported for each effect The effects of the other confounders (age, smoking, hypertension, dyslipidemia) were not statistically significant.

A MANOVA revealed a significant multivariate effect of sex and diabetes on the combined dependent variables. The univariate tests showed that sex had a significant effect on CAP, while diabetes had a significant effect on LSM. The effects of other variables (age, smoking, hypertension, dyslipidemia) were not statistically significant. We have added this information in the statistical analysis section (lines 240-242) and we have reported in the text that multivariate analysis confirmed the independent effect of sex on CAP, while diabetes on LSM (lines 329-333).

Inclusion of confounders in the logistic regression model for FLI vs. CAP (Table S3 in the Supplementary Appendix) showed that FLI retained a significant association even after adjustment for major confounders, with estimated coefficients significantly higher in men than in women. The confounders included in the model were not significant.

Table S3. Logistic Regression Model for the Association Between FLI and CAP ≥ 248 adjusted for confounders

 B

Sig.

Exp(B)

95% CI per EXP(B)

Lower

Upper

Women

FLI

.033

.007

1.034

1.011

1.057

Diabetes

.455

.386

1.576

.609

4.083

Dyslipidemia

-.287

.580

             .750

.293

1.924

SBP

.017

.292

1.018

.988

1.048

DBP

-.015

.490

.985

.948

1.024

Smoking

.102

.748

1.107

.624

1.963

Age

-.051

.162

.950

.889

1.015

Men

FLI

.109

.001

1.115

1.053

1.181

Diabetes

.843

.520

2.324

.216

22.01

Dyslipidemia

-1.820

.179

.162

.014

1.887

SBP

-.064

.101

.938

.875

1.007

DBP

-.049

.309

.953

.874

1.039

Smoking

-.763

.208

.466

.155

1.398

Age

.226

.051

1.254

1.016

1.547

SBP = systolic blood pressure; DBP= diastolic blood pressure

For women: -2 Log Likelihood = 116.754, Cox & Snell R² = 0.083, Nagelkerke R² = 0.125. For men: -2 Log Likelihood = 25.720, Cox & Snell R² = 0.381, Nagelkerke R² = 0.625.

AUC Women= 0.680 (95%CI: 0.563-0.796, p=0.005)

AUC Men= 0.920 (95%CI: 0.814-1.000), p<0.001)

 The addition of these covariates also showed an improvement in the fit parameters (e.g., -2 log likelihood and R²) and overall predictive accuracy of the model AUC in men (from 0.863 to 0.920, p<0.001), but not in women (from 0.655 to 0.680, p=0.005), confirming a greater predictive capacity in men independent of the other factors considered, as reported in lines 338-340.

Regarding the difference between the two ROC curves, we used the DeLong's test, which yielded a p-value of 0.015, indicating that the difference between the two AUCs was statistically significant. We added the sentence "To test the statistical difference between two AUC curves, the DeLong test statistic was applied (using the free statistical software R 4.5.1)" in the Statistical Analysis section (lines 246-247), and in results (lines 351-352).

Minor:

Comments 1: As for the title, the authors mention "progression," but the study is cross-sectional—no progression data. Please revise to "presentation" or "characteristics."

Response 1: We thank the reviewer for this suggestion. We agree and have changed the tile as follows: "Sex differences in MASLD after age 50: presentation, diagnosis, and clinical implications".

Comments 2: Please clarify "unsatisfactory" FLI in women—specify AUC values in the abstract.

Response 2: We thank the reviewer for this suggestion. We have added AUC values in the abstract, as suggested.

Comments 3: Tables 1-4 show some inconsistencies in reporting (e.g., age as median in text but mean in Table 1). p-values: Some borderline (e.g., BMI p=0.05)—report exact values. 4. Figure 1: Add sensitivity/specificity at optimal cut-offs.

Response 3: We thank the reviewer for this careful observation and apologize for the inconsistencies. We have revised the entire table and text as suggested. In the AUC between sexes (Figure 1), we have added sensitivity and specificity at optimal cut-offs as requested.

Comments 4: In Table 5, please clarify "B" as the beta coefficient.

Response 4: We thank the reviewer for this request. We have reported it in the table 5, as suggested.

Comments 5: All tables should add footnotes for abbreviations (e.g., PAS=PAD). And please ensure consistent units (e.g., HbA1c in % and mmol/mol).

Response 5: We thank the reviewer for this request. We have reviewed and corrected the tables as requested.

Reviewer 3 Report

Comments and Suggestions for Authors
  • In this manuscript, the authors aimed to assess sex differences in subjects with MASLD, obesity and/or T2D, aged ≥ 50 years. Despite the overall concept is interesting, this study contains several issues that need to be addressed:

Language

The English should be revised.

Abstract

The aim of the work is not obvious. Does the author want to assess differences in anthropometric and clinical features based on patient’s sex or how sex (male vs. female) influences the presence, severity, or characteristics of MASLD, obesity, and/or T2D in people over 50. It should be rewritten.

Methodology

  • In the laboratory analysis, kindly write the kits’ manufacturers and the country.
  • The ethics approval reference number and the institution that granted the approval must be specified.

Results

  • The style of table numbering should be changed as “Table n.1.” the “n.”. should be removed.
  • The significant higher number of smokers in men can act as confounding factor as it can affect the liver function.
  • You are now comparing between women and men regarding the anthropometric features. I have here many differences that can affect the comparison other than the sex. Like the stage of MASLD, diabetes, and smoking status. How did you overcome this problem?
  • Have you performed multivariant analysis?

Author Response

Reviewer 3

Comments and Suggestions for Authors

In this manuscript, the authors aimed to assess sex differences in subjects with MASLD, obesity and/or T2D, aged ≥ 50 years. Despite the overall concept is interesting, this study contains several issues that need to be addressed:

Author's reply to reviewer's report:

We sincerely thank the reviewer for carefully reading our paper. We sincerely appreciate all the valuable comments and suggestions that helped us to improve the quality of the manuscript. We have carefully reviewed the comments and addressed all concerns in the revised manuscript. Below, we provide a point-by-point response to the reviewer comments, which are highlighted in yellow in the revised manuscript.

Language

Comments 1: The English should be revised.

Response 1: We thank the reviewer for this suggestion and have revised the English as suggested.

 Abstract

Comments 2: The aim of the work is not obvious. Does the author want to assess differences in anthropometric and clinical features based on patient’s sex or how sex (male vs. female) influences the presence, severity, or characteristics of MASLD, obesity, and/or T2D in people over 50. It should be rewritten.

Response 2: We thank the reviewer for bringing this to our attention. We apologize for the lack of clarity. We have modified the aim of the abstract as follows: "[...] This observational study aimed to assess sex differences in anthropometric and clinical profiles, including non-invasive liver steatosis indices, in subjects with MASLD, obesity and/or T2D aged ≥ 50 years. [...] (lines 29-31)" and in the introduction (lines 93-94), hoping to have improved the clarity.

Methodology

Comments 3: In the laboratory analysis, kindly write the kits’ manufacturers and the country. The ethics approval reference number and the institution that granted the approval must be specified.

Response 3: We thank the reviewer for these important comments. We have added the information about the "Laboratory Analysis" section on lines 140-147 and the information about the "Ethical Statement" section on lines 224-226, as requested.

Results

Comments 4: The style of table numbering should be changed as “Table n.1.” the “n.”. should be removed.

Response 4: We thank the reviewer for this suggestion, and we have removed “n.” from all the tables, as suggested.

Comments 5: The significant higher number of smokers in men can act as confounding factor as it can affect the liver function. You are now comparing between women and men regarding the anthropometric features. I have here many differences that can affect the comparison other than the sex. Like the stage of MASLD, diabetes, and smoking status. How did you overcome this problem? Have you performed multivariant analysis?

Response 5: We sincerely thank the reviewer for this careful observation and we agree with it.

A multivariate analysis of variance (MANOVA) was performed to examine the effect of sex on the dependent variables (CAP and LSM), adjusted for potential confounders (Table S2 in the Supplementary Appendix).

Table. Multivariate analysis of variance (MANOVA) on the dependent variables (CAP and LSM). 

Effect

Test Type

Dependent Variable

F

df

p-value

Partial η²

Sex

Multivariate

3.852

2

0.023

0.036

Univariate

CAP

6.580

1

0.011

0.031

Univariate

LSM

2.036

1

0.155

0.010

Diabetes

Multivariate

4.650

2

0.011

0.043

Univariate

CAP

1.251

1

0.265

0.006

Univariate

LSM

8.844

1

0.003

0.041

F-values (F), degrees of freedom (df), significance levels (p-value), and partial eta-squared (η²) are reported for each effect. Results are adjusted for covariates (age, smoking, hypertension, dyslipidemia), none of which were statistically significant. Only significant effects are reported.

A MANOVA revealed a significant multivariate effect of sex and diabetes on the combined dependent variables. The univariate tests showed that sex had a significant effect on CAP, while diabetes had a significant effect on LSM. The effects of other variables (age, smoking, hypertension, dyslipidemia) were not statistically significant. We have added this information in the statistical analysis section (lines 240-242) and we have reported in the text that multivariate analysis confirmed the independent effect of sex on CAP, while diabetes on LSM (lines 329-333).

Inclusion of confounders in the logistic regression model for FLI vs. CAP (Table S3 in the Supplementary Appendix) showed that FLI retained a significant association even after adjustment for major confounders, with estimated coefficients significantly higher in men than in women. The confounders included in the model were not significant.

Table S3. Logistic Regression Model for the Association Between FLI and CAP ≥ 248 adjusted for confounders

 B

Sig.

Exp(B)

95% CI per EXP(B)

Lower

Upper

Women

FLI

.033

.007

1.034

1.011

1.057

Diabetes

.455

.386

1.576

.609

4.083

Dyslipidemia

-.287

.580

             .750

.293

1.924

SBP

.017

.292

1.018

.988

1.048

DBP

-.015

.490

.985

.948

1.024

Smoking

.102

.748

1.107

.624

1.963

Age

-.051

.162

.950

.889

1.015

Men

FLI

.109

.001

1.115

1.053

1.181

Diabetes

.843

.520

2.324

.216

22.01

Dyslipidemia

-1.820

.179

.162

.014

1.887

SBP

-.064

.101

.938

.875

1.007

DBP

-.049

.309

.953

.874

1.039

Smoking

-.763

.208

.466

.155

1.398

Age

.226

.051

1.254

1.016

1.547

SBP = systolic blood pressure; DBP= diastolic blood pressure

For women: -2 Log Likelihood = 116.754, Cox & Snell R² = 0.083, Nagelkerke R² = 0.125. For men: -2 Log Likelihood = 25.720, Cox & Snell R² = 0.381, Nagelkerke R² = 0.625.

AUC Women= 0.680 (95%CI: 0.563-0.796, p=0.005)

AUC Men= 0.920 (95%CI: 0.814-1.000), p<0.001)

 The addition of these covariates also showed an improvement in the fit parameters (e.g., -2 log likelihood and R²) and overall predictive accuracy of the model AUC in men (from 0.863 to 0.920, p<0.001), but not in women (from 0.655 to 0.680, p=0.005), confirming a greater predictive capacity in this subgroup independent of the other factors considered, as reported in lines 338-340.

Round 2

Reviewer 1 Report

Comments and Suggestions for Authors

The authors have done an excellent job addressing all my suggestions. Thank you for including the Strobe check list. The paper now has a clear justification for the comparison, as well as well-defined objectives.

Still a couple of minor suggestions:

In Table 5 and Suppl Table S3: For consistency and good statistical presentation, please include the leading zero for beta coefficients (e.g., “.026” should be written as “0.026”). For p-values, either style is acceptable (with or without the leading zero), but please be consistent across the table.

In Table 6: This table is key to your results. To strengthen its presentation, please provide confidence intervals (CIs) for the diagnostic accuracy estimates. This will allow readers to evaluate the precision of the reported measures.

Author Response

Reviewer 1- Round 2

Comments and Suggestions for Authors

The authors have done an excellent job addressing all my suggestions. Thank you for including the Strobe check list. The paper now has a clear justification for the comparison, as well as well-defined objectives. Still a couple of minor suggestions:

We are grateful for the reviewer's positive and thoughtful comments and sincerely appreciate the effort to further improve our manuscript. Below we provide our responses to the reviewer's comments, and the revisions are highlighted in yellow in the paper. We hope the revisions meet the reviewer's approval.

Comments 1: In Table 5 and Suppl Table S3: For consistency and good statistical presentation, please include the leading zero for beta coefficients (e.g., “.026” should be written as “0.026”). For p-values, either style is acceptable (with or without the leading zero), but please be consistent across the table.

Response 1: We thank the reviewer for this suggestion. We have changed Table 5 and Table S3 to include the leading zero for beta coefficients and for p-values, as suggested.

Comments 2: In Table 6: This table is key to your results. To strengthen its presentation, please provide confidence intervals (CIs) for the diagnostic accuracy estimates. This will allow readers to evaluate the precision of the reported measures.

Response 2: We thank the reviewer from for bringing this point to our attention. We agree with this suggestion and have added the confidence intervals (CIs) to Table 6, as suggested.

Reviewer 2 Report

Comments and Suggestions for Authors

I don't have any further comments.

Author Response

Reviewer 2- Round 2

Comments and Suggestions for Authors

I don't have any further comments.

Author’s response

We would like to thank the Reviewer for the thoughtful comments and suggestions which have greatly improved the quality of our manuscript.

Reviewer 3 Report

Comments and Suggestions for Authors

All the required comments are fulfilled

Author Response

Reviewer 3- Round 2

Comments and Suggestions for Authors

All the required comments are fulfilled

Author’s response to Reviewer’s suggestions

We appreciate the time and effort the Reviewer took to provide detailed comments on our manuscript and we are grateful for the positive response.
